# SCISSOR: Mitigating Semantic Bias through Cluster-Aware Siamese Networks for Robust Classification

**Shuo Yang**[1] **Bardh Prenkaj**[1] **Gjergji Kasneci**[1]

## Abstract

Shortcut learning undermines model generalization to out-of-distribution data. While the literature attributes shortcuts to biases in superficial features, we show that imbalances in the semantic distribution of sample embeddings induce spurious semantic correlations, compromising model robustness. To address this issue, we propose SCISSOR (Semantic Cluster Intervention for Suppressing ShORtcut), a Siamese network-based debiasing approach that remaps the semantic space by discouraging latent clusters exploited as shortcuts. Unlike prior data-debiasing approaches, SCISSOR eliminates the need for data augmentation and rewriting. We evaluate SCISSOR on 6 models across 4 benchmarks: Chest-XRay and Not-MNIST in computer vision, and GYAFC and Yelp in NLP tasks. Compared to several baselines, SCISSOR reports +5.3 absolute points in F1 score on GYAFC, +7.3 on Yelp, +7.7 on Chest-XRay, and +1 on Not-MNIST. SCISSOR is also highly advantageous for lightweight models with ∼9.5% improvement on F1 for ViT on computer vision datasets and ∼11.9% for BERT on NLP. Our study redefines the landscape of model generalization by addressing overlooked semantic biases, establishing SCISSOR as a foundational framework for mitigating shortcut learning and fostering more robust, bias-resistant AI systems.

## 1. Instruction

In recent years, machine learning models have surpassed human capabilities in various domains, such as education and E-commerce (Kasneci et al., 2023; Bodonhelyi et al., 2024). However, the high operational and usage costs of

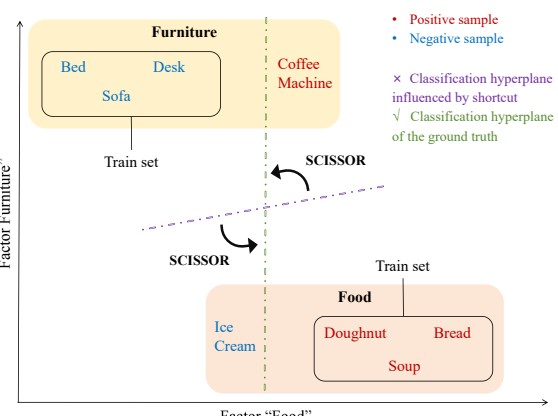

*Figure 1.* Illustration of sentiment classification, where semantic space shows clusters for "Food" and "Furniture." "Coffee Machine" and "Ice Cream" are misclassified test samples.

large language models extremely limit their scalability and practical deployment (Li & Liang, 2021; Hu et al., 2022). In contrast, the pre-training and fine-tuning pipeline offers a cost-effective and adaptable solution (Devlin et al., 2019).

However, pre-trained models often fail to maintain the performance observed during fine-tuning when applied to realistic data (Sun et al., 2024). Further analysis originated this to data biases (Yuan et al., 2024), which is known as the shortcut issue. Specifically, models rely on spurious correlations between features and labels, obtaining substantially better results on *independently-and-identically-distributed* (ID) data than on *out-of-distribution* (OOD) data.

For instance, fact-checking models may evaluate the truthfulness of a claim by counting its negations (Thorne et al., 2018), and bird models may misclassify birds as waterbirds based on water in the background (Sagawa et al., 2020). Although these shortcuts may hold for specific datasets, they significantly hinder the applicability of models to real-world scenarios (Sugawara et al., 2018).

Current research typically attributes shortcuts to the fragility of superficial features, e.g., words or pixels (Chen et al., 2023; Xu et al., 2023a), as opposed to the robustness of semantic features. However, we challenge this assumption and argue that the distribution of semantic embeddings (Reimers & Gurevych, 2019) can also appear as shortcuts.

*Equal contribution [1]Technical University of Munich, Munich, Germany. Correspondence to: Gjergji Kasneci <gjergji.kasneci@tum.de>.

*Proceedings of the 42nd International Conference on Machine Learning*, Vancouver, Canada. PMLR 267, 2025. Copyright 2025 by the author(s).

To illustrate this, consider the example in Fig. 1. Suppose we train a binary sentiment classifier using reviews from a biased E-Commerce website, where all food-related reviews are positive, and the term "Ice-cream" is not mentioned. Then, if we apply this classifier to a negative review about Ice-cream, will it classify correctly? It is unlikely, as "Ice-cream" is a food item and its word embedding is likely close to other food-related terms. Consequently, the classifier may associate the semantic region representing food with the "positive" label, thereby losing generalizability. In this case, the shortcut arises not from superficial features, but from semantic information. Similarly, in medicine, semantic shortcuts may lead to the misclassification of conditions in populations with similar physiological characteristics, resulting in fault diagnostics. As for autonomous vehicles, their systems might misinterpret ambiguous road signs due to reliance on simplified semantic patterns. All these cases highlight that semantic shortcuts may pose an urgent and previously overlooked risk.

To address these issues, we propose a novel debiasing architecture, named SCISSOR (*Semantic Cluster Intervention for Suppressing ShORtcut*), based on a Siamese network (Bromley et al., 1993).[1] Our objective is to filter out semantic information irrelevant to downstream tasks from samples exhibiting imbalanced distribution patterns. To achieve this, we first employ the *Markov Clustering Algorithm* (MCL) (Van Dongen, 2008) to cluster samples based on their semantic similarity, aiming to identify potential imbalanced areas. After that, we construct contrastive data (Chen et al., 2020) to train a debiasing module, which remaps the semantic space to disrupt the clusters that could act as shortcuts, thereby guiding the model to focus on robust features. Different from the triplet loss (Schroff et al., 2015), we consider not only the samples' original labels but also their distribution. Finally, we insert the debiasing module at the output of the pre-trained model and train it jointly with the classification head. Our contributions are as follows:

1. **Novel conceptualization of semantic bias**: To the best of our knowledge, we are the first to identify and demonstrate, both theoretically and empirically, that imbalances in the semantic distribution of samples can also lead to the shortcut problem.
2. **Lightweight, plug-and-play debiasing module**: We propose a novel debiasing approach that does not augment the training data and operates with the same time complexity as the baseline.
3. **Empirical gains across multiple domains**: We conduct experiments on text and image data using six models across text classification, style analysis, medi-

cal imaging and hand-written letter recognition tasks. Our results show that SCISSOR outperforms the baselines in terms of accuracy and F1 score.

## 2. Related Work

Over the past decade, considerable efforts have targeted the challenge of *spurious correlations*, which often undermine a model's OOD performance. Two primary lines of work relevant to our paper focus on: (1) *creating balanced and less biased datasets*, and (2) *counterfactual data generation*. In parallel, other broad techniques in *distributionally robust optimization* (DRO) or *group-based fairness* (e.g., Group DRO, IRM) typically aim to improve worst-case performance across predefined demographic groups. However, while these methods are well-suited for known protected attributes, they do not directly tackle *latent-space clustering* effects, which can give rise to semantic biases that persist even in "balanced" data. Our proposed approach, SCISSOR, is instead designed to *remap* the embedding space itself, complementing both dataset-centric and DRO-based methods by specifically targeting label-skewed clusters.

**Creating Balanced and Less Biased Datasets.** Several studies aim to address spurious correlations through data manipulation. Wu et al. (2022) propose a data generation strategy for mitigating biases in natural language inference, using a GPT-2 model with unlikelihood training to ensure label consistency and confidence-based filtering (Bartolo et al., 2021). This process identifies and discards instances that reinforce spurious correlations, thus yielding a more robust dataset. Similarly, Bras et al. (2020) use an iterative adversarial filtering approach (AFLite; Sakaguchi et al., 2020) to remove highly biased data points. For specific tasks such as fact-checking, CrossAug (Lee et al., 2021) generates negative claims and modifies evidence to create contrastive pairs, improving the model's ability to rely on genuine textual clues. Meanwhile, CLEVER (Xu et al., 2023a) attacks inference-phase biases by subtracting the output of a "claim-only" model from a more complex fusion model. Other techniques include EDA (Wei & Zou, 2019), which relies on random linguistic edits to expand training data, and "Symmetric Test Sets" (Schuster et al., 2019), which eliminates label-specific giveaways in claims. Mahabadi et al. (2020) propose "Product of Experts" to downweight spurious signals through a combination of a bias-only model with the main classifier. RAZOR (Yang et al., 2024) progressively rewrite training set through word-level feature comparison.

Although these dataset-centric strategies mitigate many surface-level or single-feature biases, they typically entail rewriting, filtering, or augmentation. Such processes dependent on careful hyperparameter tuning, and often cannot break deeper correlations *within* a pretrained model's latent

---

[1]Code available at `https://github.com/ShuoYangtum/SCISSOR`.

space. In contrast, our approach is *module-based*, directly targeting the geometry of the embedding space instead of solely relying on rewriting data. This sidesteps heavy data manipulations or repeated training on LLMs.

**Counterfactual Data Generation.** Another prevalent strategy is to produce *counterfactual* examples – perturbations of existing data designed to disentangle superficial cues from true task-relevant features. Kaushik & Lipton (2018) use manually crafted counterfactuals to demonstrate significant performance gains on challenging generalization. However, human generation can be time-consuming and often lacks diversity. Automated solutions like Polyjuice (Wu et al., 2021) and Tailor (Ross et al., 2022) fine-tune text generation models to produce specific perturbation types. While powerful, these approaches typically require *model retraining* when introducing new perturbation classes.

More recently, DISCO (Chen et al., 2023) leverages large language models to generate candidate phrasal edits, then uses a teacher model to filter out the low-quality ones. Xu et al. (2023b) adopt a similar paradigm for fact verification, generating tailored counterfactuals that reveal spurious correlations. A employs LLMs to generate counterfactual data to balance the concept of textual data. However, these approaches rely entirely on the usage of LLMs, which can be extremely resource-intensive.

While counterfactual augmentation can reduce reliance on obvious biases, it does not necessarily *eliminate* spurious semantic clusters in the embedding space. Even datasets balanced via counterfactual rewriting may contain subpopulations with label skew when projected into latent space – particularly if the pretrained model already encodes certain semantically entangled regions. Contrarily, SCISSOR directly "pulls apart" or remaps such clusters via a Siamese network head, forcing the model to focus on truly discriminative features rather than latent cluster membership.

**Why SCISSOR? Computational and Conceptual Advantages.** DRO methods also aim to ensure robustness but typically require explicit group labels or assumptions about how data sub-populations manifest. In real-world settings where biased sub-populations remain hidden or evolve over time, group-based constraints may not suffice. SCISSOR circumvents such requirements by first detecting and labeling suspicious clusters through a lightweight Markov Clustering procedure, then *disrupting* them. Additionally, SCISSOR maintains a low overhead relative to repeated data rewriting or large-scale augmentation, since it slots a debiasing module directly onto a frozen pretrained model, preserving the overall time complexity of forward passes.

Hence, our method is complementary to both data-centric and DRO-style approaches, offering a novel focus on *semantic clusters* that underlie shortcut learning. As we will show, explicitly addressing these latent clusters substantially boosts out-of-distribution performance across tasks in both NLP and computer vision.

## 3. Methodology

### 3.1. Problem Formulation

Let $\mathcal{D} = \{d_1, \ldots, d_n\}$ be a dataset containing $n$ samples and its corresponding label $y_i \in \mathcal{Y}$, where $\mathcal{Y}$ is the label set. To classify a sample $d_i$, we first transform it using a specific pre-trained embedding function $g : \mathcal{D} \to \mathcal{X} \subseteq \mathbb{R}^u$. Subsequently, we train a classifier $f_\theta : \mathcal{X} \to \mathcal{Y}$ trained by optimizing a specific loss function $\theta^* \leftarrow \arg\min_\theta \mathcal{L}(\mathcal{D}, \theta)$. Additionally, let us consider another given dataset $\mathcal{D}' = \{d'_1, \ldots, d'_m\}$ with $m$ samples drawn from a different distribution, while sharing the same label set $\mathcal{Y}$. Let us assume that the samples in $\mathcal{D}$ exhibit localized clusters with imbalanced label distributions when embedded onto the $\mathcal{X}$ space. Conversely, the distribution of samples in $\mathcal{D}'$ is relatively balanced within the same embedding space.

Considering these asusmptions, our objectives are twofold:

(1) To demonstrate the existence of semantic bias. We expect that $f_\theta$ trained on $\mathcal{X}$ to achieve better accuracy on a test set drawn from the distribution of $\mathcal{X}$ compared to one drawn from $\mathcal{X}'$. Conversely, we expect $f_\theta$, trained on $\mathcal{X}'$ to exhibit comparable performance on the test sets drawn from both $\mathcal{X}$, and $\mathcal{X}'$.

(2) To enhance the performance of $f_\theta$ trained on $\mathcal{X}$ on test sets drawn from $\mathcal{X}'$ through semantic debiasing algorithms.

### 3.2. Demonstration of Semantic Bias Existence

To elucidate the underlying causes and influential factors of semantic shortcuts, we propose the following lemmas. Lemma 3.1 quantifies how small changes in the representation space influence the classifier's outputs, revealing the extent to which the model's sensitivity to input variations may contribute to semantic bias.

**Lemma 3.1.** *Given a differentiable classifier $f_\theta$, two input samples $d, d' \in \mathcal{D}$, a function $g : \mathcal{D} \to \mathbb{R}^u$, if the Euclidean distance between $g(d)$ and $g(d')$ in a $u$-dimensional space is lower than $\alpha$, then the Euclidean distance between their outputs given from $f_\theta$ is upper-bounded by*

$$\sqrt{d} \cdot \alpha \cdot \|\nabla f_\theta(g(d))\|_2 + \frac{1}{2}M\alpha, \qquad (1)$$

*where $M$ is the upper bound of the norm of the Hessian matrix of $f_\theta$ over the segment connecting $g(d)$ with $g(d')$.*

Lemma 3.2 formalizes how local imbalances in training labels lead to biased classifier behavior, showing that majority-label dominance causes systematic misclassification of

minority-labeled samples, with this effect worsening as training progresses.

**Lemma 3.2.** *Let $f_\theta$ be a differentiable classifier trained on data embedded into $\mathcal{X} \subseteq \mathbb{R}^u$ by a function $g$. Fix an anchor point $c \in \mathcal{X}$ and a radius $\alpha > 0$. Suppose that within the ball $B(c, \alpha) = \{x \in \mathcal{X} : ||x - c|| < \alpha\}$, the training labels are imbalanced–i.e., one majority label is heavily represented compared to a minority label. Then:*

*(1) any classifier that minimizes empirical risk will tend to predict the majority label for most points in $B(c, \alpha)$;*

*(2) the expected misclassification probability on the minority-labeled samples in $B(c, \alpha)$ is bounded below by a positive constant;*

*(3) the bound increases over the course of training.*

We refer the reader to Appendix B for the omitted proofs, and to Appendix C for a detailed discussion about the significance of our theoretical findings in semantic biases.

### 3.2.1. THEORY-GROUNDED OBSERVATIONS

*Imbalance-Induced Misclassification.* As a classifier becomes increasingly accurate on the majority-labeled samples (e.g., "positives") in a given region, the lower bound on misclassification for minority-labeled samples (e.g., "negatives") in the same region grows. In other words, once the model effectively learns to recognize the majority label, any semantically similar minority samples become more likely to be wrongly classified as majority

*Concentration Boosts Shortcut Misclassification.* As $\alpha \rightarrow 0$ the data around an anchor point becomes more tightly concentrated. Under label imbalance, this yields a higher lower bound on the expected misclassification rate for the minority label. In other words, the more localized and imbalanced the samples are, the more the classifier relies on a shortcut that favors the majority label—making minority misclassification increasingly inevitable.

*Training Exacerbates Shortcut Misclassification.* As model parameters converge during training, the lower bound on the expected misclassification probability for minority-labeled samples rises. This indicates that shortcut-based misclassifications intensify over the course of training, further harming minority classes.

*Larger Models Reduce Shortcut Misclassification.* If the network's capacity increases—reflected by a higher Hessian norm bound $M$ or a larger embedding dimension $u$—the theoretical lower bound on the expected misclassification probability for minority samples decreases. Consequently, deeper or more expressive models are more resilient against shortcut-based misclassifications than smaller models.

### 3.3. Markov Clustering

To determine the initial distribution of the given samples $\mathcal{D}$, we apply the MCL to their embeddings $\mathcal{X}$, which includes three steps.

*Constructing the Markov matrix.* We compute the cosine similarity between the embeddings to form a similarity matrix, which is then normalized to generate a Markov matrix (Kelly, 1981). Each entry in the matrix represents the transition probabilities between two samples.

*Expansion and inflation.* We square the matrix to simulate the probability distribution after random walks, propagating and expanding the connections between samples. Subsequently, we raise each matrix element to its power and normalize again to emphasize stronger connections and suppress weaker ones.

*Convergence and clustering.* By repeating the expansion and inflation, the matrix will converge into a sparse block-diagonal structure, where each block represents a cluster. Finally, we assign samples to different clusters accordingly. For each cluster, we categorize it into two groups based on the label distribution of its samples: i.e., balanced and imbalanced clusters. As discussed in Lemma 3.2, the semantics of samples within imbalanced clusters can introduce shortcut learning, hence reducing model generalizability. Therefore, we argue that samples from imbalanced clusters form $\mathcal{X}$ and those from balanced ones form the $\mathcal{X}'$. Our goal is to mitigate semantic biases present in $\mathcal{X}$ to enhance the performance of $f_\theta$ on $\mathcal{X}'$.

Additionally, we account for the unequal number of samples within each cluster group. To prevent potential biases caused by sample imbalance, we perform random downsampling such that the number of samples in the two cluster groups is equal, and both groups maintain a balanced label distribution in total. In Appendix E, we discuss on the scalability and time complexity of the MCL.

### 3.4. Semantic Debiasing

We propose a plug-and-play debiasing module designed to filter out classification-irrelevant semantic features. We train and integrate a lightweight neural network to the output of a pre-trained language model (PLM), remapping its embedding space (Devlin et al., 2019).

**Construction of contrastive data.** Here, we present a novel idea for constructing contrastive samples with consideration of their semantic distribution. Unlike conventional approaches that solely maximize the distance between samples with different labels (Shah et al., 2022), we disrupt the clustering tendencies within the samples that can serve as shortcuts. Therefore, we introduce the concept of an "intermediate sample." For a given anchor, positive samples share

**Step 1** Clustering and create a quadruplet.    **Step 2** Creating Triplets and train a debiasing module.    **Step 3** Freeze the debiasing module and train a classification head.

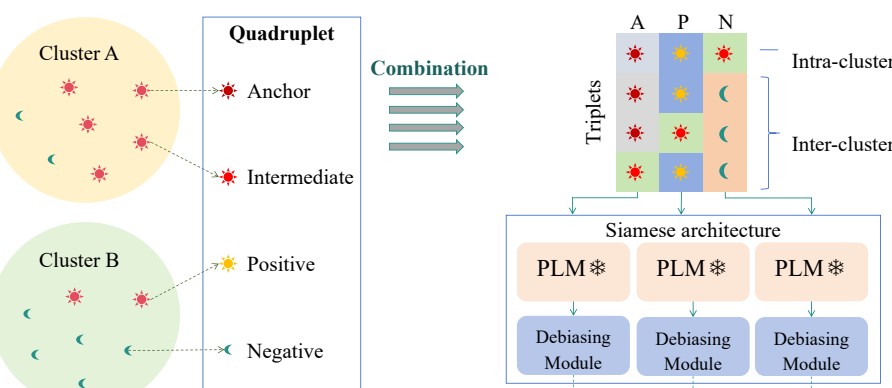
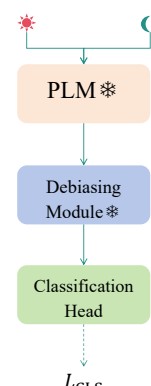

*Figure 2.* **Overview of SCISSOR.** The "Sun" and "Moon" symbols represent samples with two different labels. We first apply the clustering algorithm to group samples based on the similarity of their embeddings, identifying clusters with label imbalances. Next, we generate triplets (Positive, Intermediate, Negative) by selecting an anchor sample based on their semantic clustering properties. Their purpose is to guide (train) the debiasing module in remapping the embedding space of the PLM by discouraging shortcut-induced cluster formations, ultimately improving classifier robustness.

the same label but belong to different clusters. Intermediate samples share the same label and cluster as the anchor. Negative samples have a different label. The intermediate sample plays a dual role: (1) it acts as a negative sample when contrasted with a positive one, and (2) as a positive sample when contrasted with a negative one. Any anchor in $\mathcal{X}$ forms a training quadruplet with its corresponding positive, intermediate and negative samples.

*Table 1.* Four ways to create triplets from a quadruplet. "A", "P", "I" and "N" stand for anchor, positive, intermediate and negative sample in a quadruplet, respectively.

| Triplet factor | Anchor | Positive | Negative |
|---|---|---|---|
| | A | P | N |
| Inter-cluster | A | I | N |
| | I | P | N |
| Intra-cluster | A | P | I |

Therefore, for each quadruplet, there are $\binom{4}{3} = 4$ possible ways to decompose it into triplets of the form (anchor, positive, negative), as shown in Table 1. Specifically, for each anchor, we employ two methods to construct a triplet.

*(1) Inter-cluster Contrast.* Divide positive and negative samples in the triplet based on their labels, to encourage samples with different labels to be further apart in the semantic space.

*(2) Intra-cluster Contrast.* Define positive and negative samples in triplets based on clustering characteristics. This promotes differentiation among semantically similar samples within the same class, preventing the model from learning

shortcuts from classification-irrelevant features.

We then rely on the triplet loss to train the debiasing module.

$$\mathcal{L} = \sum_{i=1}^{n} \max(0, \cos(a_i, p_i) - \cos(a_i, n_i) + \beta), \quad (2)$$

where $a_i, p_i$, and $n_i$ represent the anchor, positive, and negative samples in the $i$-th triplet, $\cos(\cdot, \cdot)$ denotes the cosine distance function, and $\beta$ is the margin ensuring the distance between the anchor and the negative sample exceeds that between the anchor and the positive sample. In this setup, all three samples are simultaneously fed into a shared-parameter Siamese network, where the goal is to maximize the distance between $a_i$ and $n_i$ while minimizing that between $a_i$ and $p_i$.

After training, we implement the debiasing module by inserting it at the output of a frozen PLM. This approach offers two key advantages: (1) high adaptability, and (2) low resource consumption. In other words, the debiasing module can seamlessly integrate with any architecture without modifying the PLMs. Moreover, the time complexity of our lightweight debiasing network is linear, ensuring that the entire complexity is dominated by the PLM usage. Finally, SCISSOR does not rely on data augmentation (Lee et al., 2021) or LLM distillation (Yang et al., 2024), distinguishing it from other token- and pixel-level debiasing methods.

**Alternating Training with Clustering and Remapping.** Lastly, we employ an alternating training strategy between clustering and debasing. We argue that, as sample embedding distributions update, their cluster assignments change

dynamically. Samples that move out from their original clusters may inadvertently align with others, forming new imbalanced clusters. To prevent this, we alternately implement the clustering and the debiasing. Fig 2 illustrates our method.

We train the module until the samples within the imbalanced cluster no longer exhibit clustering tendencies in the remapped semantic space. To quantify the changes in clustering behavior, we employ the Hopkins Statistic. During training, this metric gradually increases and stabilizes near 0.5, indicating the removal of clustering tendencies. Finally, we freeze the parameters of the debiasing module and train a classification head with the cross-entropy loss $\mathcal{L}_{\text{CLS}}$.

## 4. Experiments

### 4.1. Datasets

We evaluate SCISSOR across four classification tasks: letter recognition, medical test, sentiment classification and style analysis. These tasks were conducted on two computer vision datasets and two natural language processing datasets.

**Computer Vision (CV).** The Not-MNIST is a multi-label classification dataset (Bulatov, 2011) contains 19,000 images of hand-written letters. The Chest-XRay dataset (Hagos et al., 2023) contains 5,863 XRay images depicting both healthy lungs (0) and lungs affected by pneumonia (1).

**Natural Language Processing (NLP).** The Yelp dataset consists of user reviews (positive and negative) of various businesses on Yelp. We use the version presented in (Dai et al., 2019). The Grammarly's Yahoo Answers Formality Corpus (GYAFC) (Rao & Tetreault, 2018) is the largest dataset for style transfer, containing 110,000 informal and formal sentence pairs.

Note that the shortcuts we identified have not been previously reported in the literature. Thus, no standard adversarial datasets are currently available for robustness evaluation. To address this limitation, we adopted the cross-validation method shown in (Yang et al., 2024).

### 4.2. Experimental Setup

We use six models as baseline classifiers $f_\theta$: BERT (Devlin et al., 2019), RoBERTa (Liu et al., 2019), LLaMA 3.2 (Meta, 2024), Vision Transformer (ViT) (Dosovitskiy et al., 2020), Shifted Window Hierarchical Visio e nansformer (Swin) (Liu et al., 2021), and DINOv2 (Oquab et al., 2024).[2] We employ the AdamW optimizer (Loshchilov & Hutter, 2019) with an initial learning rate of $3 \times 10^{-5}$ and trained the models with batch size equal to 8 across four

NVIDIA A100 Tensor Core-GPUs. Our debiasing network consists of a single Transformer module, which includes an attention layer and a feedforward layer, with 8 attention heads and 768 neurons per layer. Our classification head is a single linear layer with 768 neurons. The value of $\beta$ in Equation 2 is 0.2. To simplify training, we construct a quadruplet for each anchor using random sampling.

### 4.3. Results

**Validation of Semantic Shortcuts.** To experimentally validate the theoretical findings presented, we first compared the robustness of classifiers trained on balanced cluster data and imbalanced cluster data. These differences can be measured by evaluating the classifiers' performance on ID test sets versus OOD test sets. We report the classification accuracy and F1 scores in Fig. 3. Given that imbalanced clusters exhibit pronounced shortcut features, we treated them as the optimization target and the balanced clusters as test data representing real-world scenarios in the subsequent experiments. To further illustrate the inherent clustering structure of the datasets in the semantic space, we calculated the Hopkins statistic for these datasets after being embedded by the initial model. Tables 2 and 3 show these statistics for the NLP and CV datasets, respectivelz.

**Effectiveness of the Proposed Method.** To evaluate the debiasing capability of SCISSOR, we evaluate its gain on accuracy and F1 score over the baseline classifiers. We compare SCISSOR against three state-of-the-art debiasing methods: RAZOR (Yang et al., 2024) for NLP tasks, LC (Liu et al., 2023) for CV tasks and IRM (Arjovsky et al., 2019) for both tasks. Specifically, RAZOR relies on rewriting training data containing potential biases using LLMs, while LC corrects classifier logits to balance the gradient impact of majority and minority groups during training. Tables 4 and 5 illustrate the results.

*Table 2.* Hopkins Statistic of NLP datasets. Low values indicate stronger clustering tendency.

|  | BERT | RoBERTa | LLaMA |
|---|---|---|---|
| GYAFC | $1.59 \times 10^{-8}$ | $5.42 \times 10^{-8}$ | 0 |
| Yelp | $1.74 \times 10^{-8}$ | $5.59 \times 10^{-8}$ | $3.34 \times 10^{-4}$ |

*Table 3.* Hopkins Statistic of CV datasets. Low values indicate stronger clustering tendency.

|  | ViT | Swin | DINOv2 |
|---|---|---|---|
| Chest-XRay | $8.87 \times 10^{-7}$ | $7.83 \times 10^{-9}$ | $5.06 \times 10^{-8}$ |
| Not-MNIST | $4.07 \times 10^{-7}$ | $4.93 \times 10^{-9}$ | $3.01 \times 10^{-8}$ |

---

[2]All models are publicly available at `https://huggingface.co`.

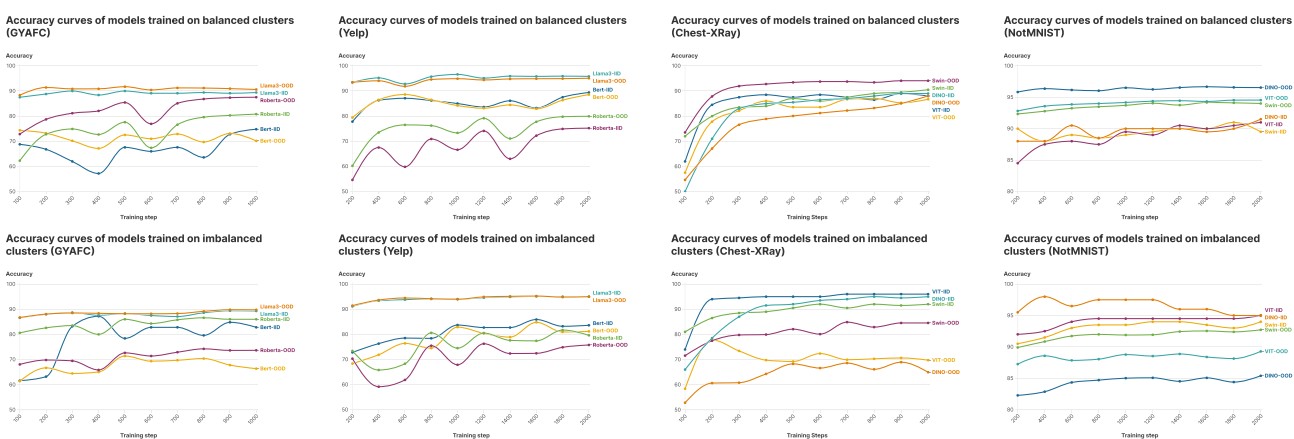

*Figure 3.* **Models trained on imbalanced clusters exhibit significant performance drops on OOD data, confirming that semantic bias harms generalization, while balanced training improves robustness.** We show the performance of classifiers on balanced and imbalanced clusters under ID and OOD test data. For each data group, we randomly choose 500 from the training set to form the test set.

*Table 4.* SCISSOR's impact on the baseline classifiers on the NLP datasets in terms of Accuracy and F1 score. Bold values indicate best performing per baseline; underlined the second-best.

|  | GYAFC | | Yelp | |
|---|---|---|---|---|
|  | ACC ↑ | F1 ↑ | ACC ↑ | F1 ↑ |
| BERT | 70.40 | 0.70 | 81.17 | 0.81 |
| BERT (w/ IRM) | 77.82 | 0.77 | 87.29 | 0.87 |
| BERT (w/ RAZOR) | 72.76 | 0.71 | 82.20 | 0.82 |
| BERT (w/ SCISSOR) | **78.20** | **0.78** | **90.65** | **0.91** |
| RoBERTa | 73.66 | 0.73 | 75.76 | 0.76 |
| RoBERTa (w/ IRM) | 79.61 | 0.79 | 84.53 | 0.84 |
| RoBERTa (w/ RAZOR) | 73.58 | 0.73 | 76.36 | 0.76 |
| RoBERTa (w/ SCISSOR) | **81.34** | **0.81** | **87.79** | **0.88** |
| LLaMA | 89.40 | 0.89 | 95.00 | 0.95 |
| LLaMA (w/ IRM) | 83.65 | 0.83 | 94.87 | 0.95 |
| LLaMA (w/ RAZOR) | 87.00 | 0.86 | 94.44 | 0.94 |
| LLaMA (w/ SCISSOR) | **89.46** | **0.89** | **95.20** | **0.95** |

*Table 5.* SCISSOR's impact on the baseline classifiers on the CV datasets in terms of Accuracy and F1 score. Bold values indicate the best performing per baseline; underlined the second-best.

|  | Chest-XRay | | Not-MNIST | |
|---|---|---|---|---|
|  | ACC ↑ | F1 ↑ | ACC ↑ | F1 ↑ |
| ViT | 72.38 | 0.72 | 88.87 | 0.89 |
| ViT (w/ IRM) | 80.47 | 0.80 | 89.37 | 0.89 |
| ViT (w/ LC) | 82.73 | 0.82 | 89.25 | 0.89 |
| ViT (w/ SCISSOR) | **83.92** | **0.84** | **90.89** | **0.91** |
| Swin | 84.54 | 0.84 | 92.72 | 0.93 |
| Swin (w/ IRM) | 76.92 | 0.76 | 92.12 | 0.92 |
| Swin (w/ LC) | 79.75 | 0.79 | 91.56 | 0.92 |
| Swin (w/ SCISSOR) | **88.65** | **0.89** | **92.74** | **0.93** |
| DINOv2 | 68.94 | 0.66 | 85.40 | 0.85 |
| DINOv2 (w/ IRM) | 69.64 | 0.67 | 85.39 | 0.85 |
| DINOv2 (w/ LC) | 72.73 | **0.72** | 83.63 | 0.84 |
| DINOv2 (w/ SCISSOR) | **73.59** | **0.72** | **85.75** | **0.86** |

## 4.4. Analysis and Discussion

**Larger models exhibit up to $\times 10^3$ higher Hopkins statistics, revealing a weaker cluster effect in embeddings.** From the Hopkins statistics reported in Tables 2 and 3, we observe that all models produce values close to 0 across every dataset. Nonetheless, Yelp and GYAFC share relatively similar clustering tendencies, whereas Not-MNIST exhibits markedly higher randomness compared to Chest-XRay. This finding suggests that natural data tends to exhibit a strong inherent clustering structure within the embedding space of pretrained language models (PLMs). Such an observation provides theoretical support for SCISSOR's approach of assigning cluster-based labels in this space.

Additionally, we note that the Hopkins statistic monotonically increases with model size. Smaller networks yield more concentrated embedding distributions for the same

*Table 6.* Adjusted rand index between topics and semantic clusters.

|  | BERT | RoBERTa | LLaMA |
|---|---|---|---|
| GYAFC | 0.62 | 0.13 | 0.00 |
| Yelp | 1.93 | 2.44 | 0.15 |

dataset, while larger models like LLaMA display less concentration. In vision models, ViT and DINOv2 similarly show Hopkins statistics that are one to two orders of magnitude higher than Swin, consistent with the trend that larger parameter counts lead to higher Hopkins values.

**Semantic Imbalance Causes Up to 20-Point OOD Accuracy Drops, While Balanced Training Enhances Robustness.** Under in-distribution (ID) conditions, all models achieve high accuracy (Fig. 3). However, when trained on semantically imbalanced datasets, their performance substantially degrades on out-of-distribution (OOD) test sets.

*Table 7.* Ablation study on the NLP datasets in terms of Accuracy and F1 score with K-means clustering.

| | GYAFC | | Yelp | |
|---|---|---|---|---|
| | ACC ↑ | F1 ↑ | ACC ↑ | F1 ↑ |
| BERT | 70.40 | 0.70 | 81.17 | 0.81 |
| BERT (w/ RAZOR) | 72.76 | 0.71 | 82.20 | 0.82 |
| BERT (w/ Triplet) | 77.60 | 0.77 | 88.60 | 0.89 |
| BERT (w/ SCISSOR, k-means) | 76.52 | 0.75 | 88.50 | 0.89 |
| BERT (w/ SCISSOR) | **78.20** | **0.78** | **90.65** | **0.91** |
| RoBERTa | 73.66 | 0.73 | 75.76 | 0.76 |
| RoBERTa (w/ RAZOR) | 73.58 | 0.73 | 76.36 | 0.76 |
| RoBERTa (w/ Triplet) | 81.32 | 0.81 | 87.70 | 0.87 |
| RoBERTa (w/ SCISSOR, k-means) | 81.30 | 0.81 | 87.78 | 0.87 |
| RoBERTa (w/ SCISSOR) | **81.34** | **0.81** | **87.79** | **0.88** |
| LLaMA | 89.40 | 0.89 | 95.00 | 0.95 |
| LLaMA (w/ RAZOR) | 87.00 | 0.86 | 94.44 | 0.94 |
| LLaMA (w/ Triplet) | 89.40 | 0.89 | 94.64 | 0.94 |
| LLaMA (w/ SCISSOR, k-means) | 89.42 | 0.89 | 94.90 | 0.95 |
| LLaMA (w/ SCISSOR) | **89.46** | **0.89** | **95.20** | **0.95** |

This underscores a robustness gap driven by shortcut learning. In computer vision (CV), the largest OOD accuracy drop occurs on Chest-XRay—ViT and DINOv2 each lose 20 points, and even Swin experiences a 2-point decline on Not-MNIST. By contrast, when training data exhibits balanced semantic clusters, there is no consistent ID-OOD performance gap, indicating a higher degree of robustness.

A similar pattern emerges in textual datasets, with the shortcut effect most pronounced on GYAFC. Among the language models tested, BERT—when trained on imbalanced clusters—shows the greatest performance gap ( 20 points) between ID and OOD. However, when BERT is trained on balanced data, the gap narrows to 3 points. Notably, LLaMA achieves nearly identical results on both ID and OOD tests. As shown in Table 2, LLaMA exhibits strong overall performance and lower embedding concentration, making it less prone to shortcut-driven errors.

**SCISSOR Achieves Up to 12-Point Gains in Accuracy and F1 Across NLP and Vision Tasks.** Tables 4 and 5 summarize the performance improvements introduced by SCISSOR. In the NLP domain (Table 4), SCISSOR delivers notable gains for all three language models. On GYAFC, for instance, BERT and RoBERTa each see a 7-point boost in accuracy and F1 score relative to the RAZOR baseline, while on Yelp, SCISSOR outperforms both BERT and RoBERTa by 9 and 12 points, respectively. Although LLaMA already exhibits robust performance against shortcuts, it still realizes marginal benefits from SCISSOR. Moreover, because the datasets in question are label-balanced and contain limited superficial shortcuts, RAZOR's strategy of manipulating superficial features and data rewriting actually lowers

LLaMA's accuracy by ∼2 points.

In computer vision, SCISSOR's largest gains occur on Chest-XRay, where ViT achieves a 12-point increase in both accuracy and F1 on the OOD set. Across the board, SCISSOR consistently outperforms LC in terms of both accuracy and F1. We attribute LC's shortfall to its inability to address deeper semantic biases in balanced-label scenarios, thereby limiting its improvement potential. Performance improvements on Not-MNIST are relatively smaller, likely due to the dataset's weaker embedding clusters and lower susceptibility to shortcut issues; even so, SCISSOR still provides a 2-point lift in accuracy and F1 for ViT.

We observe that although IRM mitigatesrtcuts in many cases, our method still significantly outperforms it across all tests. Moreover, IRM performs worse than the baseline on small datasets, such as Chest-XRay (w/ SWIN) and GYAFC (w/ LLaMA). We attribute this to IRM assigning excessive training weight to features that remain invariant, which prevents other useful features from being accurately identified and utilized.

### 4.4.1. WHY DO SEMANTIC CLUSTERS MATTER?

While our experiments demonstrate that debiasing semantic clusters improves generalization, one might still ask: *What do these clusters actually represent in practice?* To investigate, we hypothesize that samples within the same cluster tend to share common semantic themes or topics. To test this, we trained a Latent Dirichlet Allocation (LDA) topic model (Blei et al., 2003) and measured the alignment between semantic clusters and topic clusters using the Adjusted Rand Index (ARI) (Hubert & Arabie, 1985). The

*Table 8.* Ablation study on the CV datasets in terms of Accuracy and F1 score.

| | Chest-XRay | | Not-MNIST | |
|---|---|---|---|---|
| | ACC ↑ | F1 ↑ | ACC ↑ | F1 ↑ |
| ViT | 72.38 | 0.72 | 88.87 | 0.89 |
| ViT (w/ LC) | 82.73 | 0.82 | 89.25 | 0.89 |
| ViT (w/ Triplet) | 82.12 | 0.81 | 88.97 | 0.89 |
| ViT (w/ SCISSOR, k-means) | 82.20 | 0.82 | 89.44 | 0.90 |
| ViT (w/ SCISSOR) | **83.92** | **0.84** | **90.89** | **0.91** |
| Swin | 84.54 | 0.84 | 92.72 | 0.93 |
| Swin (w/ LC) | 79.75 | 0.79 | 91.56 | 0.92 |
| Swin (w/ Triplet) | 86.64 | 0.86 | 91.00 | 0.91 |
| Swin (w/ SCISSOR, k-means) | 87.58 | 0.87 | 92.72 | 0.93 |
| Swin (w/ SCISSOR) | **88.65** | **0.89** | **92.74** | **0.93** |
| DINOv2 | 68.94 | 0.66 | 85.40 | 0.85 |
| DINOv2 (w/ LC) | 72.73 | 0.72 | 83.63 | 0.84 |
| DINOv2 (w/ Triplet) | 72.88 | 0.72 | 84.58 | 0.85 |
| DINOv2 (w/ SCISSOR, k-means) | 72.80 | 0.72 | 85.70 | 0.85 |
| DINOv2 (w/ SCISSOR) | **73.59** | **0.72** | **85.75** | **0.86** |

results, shown in Table 6, reveal a clear positive correlation between semantic clustering and topic clustering across all datasets and models. This strongly suggests that semantic clusters are not just artifacts of model embeddings—they capture meaningful, high-level concepts in the data. An intriguing insight emerges from our findings: *stronger models, such as LLaMA, exhibit significantly lower ARI scores*. This implies that as models grow in capacity, their embeddings become less tightly coupled to discrete topics. In other words, more powerful models learn richer, more distributed representations, rather than rigidly grouping samples by surface-level themes. This phenomenon aligns with our earlier observation that larger models are naturally more resistant to shortcut learning—a key insight into why SCISSOR has a greater impact on smaller architectures. These findings provide compelling evidence that shortcut learning is fundamentally tied to how models organize semantic information, and that disrupting these clusters can lead to more robust, generalizable classifiers.

### 4.5. Ablation Study

We investigated the impact of clustering algorithms on the effectiveness of SCISSOR. Specifically, we replaced MCL with the K-means clustering algorithm and repeated the comparative experiments with the baselines, as shown in Table 7 and 8.

We observed that the clustering algorithm cannot significantly impact the effectiveness of SCISSOR. After replacing MCL with K-means, which holds a linear time complexity with respect to data scale, our approach showed almost identical performance in Accuracy and F1 scores while maintaining a significant advantage over the baselines.

Additionally, compared to Triplet, SCISSOR consistently demonstrates a advantage about 2 points. We analyze that this is because Triplet focuses solely on optimizing samples based on their classification labels, neglecting the importance of the embedding distribution. During the training process of Triplet, samples with the same label are pulled closer together, which could lead to the formation of new imbalanced semantic clusters.

## 5. Conclusion

Shortcut learning remains a fundamental challenge in machine learning, undermining model generalization by encouraging reliance on spurious correlations. While prior work has focused primarily on surface-level biases, we reveal that semantic clustering effects within embedding spaces also contribute significantly to shortcut-driven failures. To address this, we introduce SCISSOR, a novel cluster-aware Siamese network that actively disrupts latent semantic shortcuts without requiring data augmentation or rewriting. Our extensive experiments across six models and four benchmarks demonstrate that SCISSOR substantially improves out-of-distribution robustness, particularly in settings where traditional debiasing techniques struggle. Notably, SCISSOR achieves +7.7 F1 points on Chest-XRay, +7.3 on Yelp, and +5.3 on GYAFC, setting a new standard for mitigating shortcut learning at the semantic level. Furthermore, we show that larger models are naturally more resistant to these biases, yet SCISSOR enhances even smaller models, making them competitive with larger, more complex architectures.

## Impact Statement

This paper presents work whose goal is to advance the field of Machine Learning. There are many potential societal consequences of our work, none which we feel must be specifically highlighted here.

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

# A. More Theoretical Study

## A.1. Why Can SCISSOR Reduce Generalization Error?

Typically, the generalization error is:

$$\mathbb{E}_{D_{\text{test}}}[L(f_\theta(x), y)] - \mathbb{E}_{D_{\text{train}}}[L(f_\theta(x), y)], \tag{3}$$

where $\mathbb{E}_{D_{\text{test}}}[L(f_\theta(x), y)]$ is the loss on the OOD test set and $\mathbb{E}_{D_{\text{test}}}[L(f_\theta(x), y)]$ is the loss on during training.

However, it is challenging to pre-define the distribution patterns of OOD data. Therefore, we use Rademacher Complexity to measure the fitting ability of SCISSOR on random noise, which in turn reflects its generalization capability. The Rademacher Complexity is define by:

$$\hat{\mathcal{R}}_S(\mathcal{F}) = \mathbb{E}_\sigma \left[ \sup_{f \in \mathcal{F}} \frac{1}{m} \sum_{i=1}^{m} \sigma_i f(x_i) \right], \tag{4}$$

where $\sup_{f \in \mathcal{F}}$ represents finding the function $f$ within the hypothesis class $\mathcal{F}$ that maximizes the Rademacher Complexity. $\sigma_i$ is the Rademacher variable, ranging from $[1, -1]$.

Theoretically, SCISSOR increases the intra-cluster contrastive loss, which filters out semantic information related to clustering. As a result, it expands the effective dimensionality of the data. This enables the model to learn more robust, classification-relevant features rather than relying on unstable semantic shortcuts, thereby reducing its ability to fit noise. Consequently, the Rademacher complexity of $\mathcal{F}$ decreases, indicating stronger generalization ability, which in turn reduces generalization error.

## A.2. Why Is SCISSOR's Classification Hyperplane More Robust?

To further illustrate how SCISSOR remap the semantic space structure to support the building of the classification boundary, we introduce convex geometric analysis. We assume that a sample embedding $x$ corresponds to a specific label within the label set $\mathcal{Y} = \{y_1, \ldots, y_N\}$ and that samples of class $y_i$ are distributed within the label spherical cluster $C_i$. Typically, the classifier $\theta$ performs precise classification by separating these label spherical clusters. However, when $x$ is unevenly distributed, it difficult to establish an accurate classification plane.

We assume that the spherical centers of any cluster $C_i$ is given by $\mu_i$:

$$\mu_i = \mathbb{E}_{x \sim C_i} x, \tag{5}$$

the inter-class separation can be measured using cosine similarity:

$$cos(\theta) = \frac{\sum_{i=0}^{N} \sum_{j=0, i \neq j}^{N} \frac{<\mu_i, \mu_j>}{\|\mu_i\| \|\mu_j\|}}{\binom{N}{2}}. \tag{6}$$

The larger the $cos(\theta)$, the higher the degree of overlap between the class boundaries, making it difficult to establish an accurate classification plane. We calculated change in $cos(\theta)$, denoted as $\Delta\theta$, before and after SCISSOR training, i.e. $\theta_{\text{before}}$ and $\theta_{\text{after}}$, to demonstrate how SCISSOR increases the distance between samples with different labels:

$$\Delta\theta = cos(\theta_{\text{before}} - \theta_{\text{after}}) \tag{7}$$

The results is shown in Table 9 and 10. We observed that by using SCISSOR, the distance between different label samples increased for all models across all datasets.

*Table 9.* $\Delta\theta$ of NLP datasets. Positive values indicate that the distance between samples with different labels increased.

|  | BERT | RoBERTa | LLaMA |
|---|---|---|---|
| GYAFC | 0.40 | 0.40 | 0.02 |
| Yelp | 0.26 | 0.19 | 0.01 |

*Table 10.* $\Delta\theta$ of CV datasets. Positive values indicate that the distance between samples with different labels increased.

|  | ViT | Swin | DINOv2 |
|---|---|---|---|
| Chest-XRay | 0.35 | 0.22 | 0.32 |
| Not-MNIST | 0.16 | 0.05 | 0.17 |

## B. Omitted Proofs

### B.1. Lemma 1

*Proof.* Given $\|g(d) - g(d')\|_2 < \alpha$, according to the Cauchy–Schwarz inequality (Garibaldi et al., 2007), the following holds for $\forall v$:

$$\|v\| = \sum_{i=1}^{u} |v| \leq \left( \sum_{i=1}^{u} v_i^2 \right)^{\frac{1}{2}} = \sqrt{u} \cdot \|v\|_2. \tag{8}$$

Let $v = g(d) - g(d')$, then:

$$\|g(d) - g(d')\| \leq \sqrt{u} \cdot \|g(d) - g(d')\|_2 < \alpha\sqrt{u}. \tag{9}$$

Since $f_\theta$ is differentiable, we apply its Taylor expansion (Taylor, 1715) as follows:

$$f_\theta(g(d')) = f_\theta(g(d)) + \nabla f_\theta(g(d))^T (g(d) - g(d')) + R, \tag{10}$$

where $R$ represents the higher-order terms, satisfying:

$$R = \frac{1}{2}(g(d') - g(d))^T H f_\theta(\xi)(g(d') - g(d)), \tag{11}$$

where, $H f_\theta(\xi)$ is the Hessian matrix (Hemati et al., 2023) of $f_\theta$ with $\xi \in [g(d), g(d')]$.

Assuming that there is no gradient explosion issue in $f_\theta$, it has a finite number of parameters, and all parameters are bounded, then $H$ is continuous, and its norm has an upper bound $M$:

$$\|H f_\theta(\xi)\|_2 \leq M, \forall \xi \in [g(d), g(d')] \tag{12}$$

Here,

$$\|R\| \leq \frac{1}{2}M\|g(d) - g(d')\|_2 < \frac{1}{2}M\alpha. \tag{13}$$

Substituting it into Eq. (10), we have:

$$\begin{aligned} \|f(g(d)) - f(g(d'))\|_2 &\leq \|f(g(d)) - f(g(d'))\| \\ &< \|\nabla f_\theta(g(d))\| \cdot \|g(d) - g(d')\| + \frac{1}{2}M\alpha \\ &\leq \sqrt{u} \cdot \alpha \cdot \|\nabla f_\theta(g(d))\|_2 + \frac{1}{2}M\alpha \end{aligned} \tag{14}$$

$\square$

## B.2. Lemma 2

*Proof.* Let us consider a multi-label classification, within the neighborhood of $c$, given a majority sample set $\mathcal{D} = (d_1, ..., id_{N_1})$ with corresponding label $y_{\text{major}}$, and any minority sample set $\hat{\mathcal{D}} = (\hat{d}_1, ..., \hat{d}_{N_2})$ with corresponding label $y_{\text{minor}}$. We suppose $N_1 \gg N_2$, and the output of $f_\theta$ is the probability of a sample obtaining label $y_{\text{major}}$, that is $f_\theta(g(d)) = P(y_{\text{major}}|x; \theta)$. By applying the Lemma 1, we have

$$\begin{aligned} \|P(y|c;\theta) - P(y|d;\theta)\|_2 &\leq \|P(y|c;\theta) - P(y|d;\theta)\| \\ &< \sqrt{u} \cdot \alpha \cdot \|\nabla f_\theta(g(c))\|_2 + \frac{1}{2}M\alpha, \end{aligned} \tag{15}$$

and

$$\begin{aligned} \|P(y|c;\theta) - P(y|\hat{d};\theta)\|_2 &\leq \|P(y|c;\theta) - P(y|\hat{d};\theta)\| \\ &< \sqrt{u} \cdot \alpha \cdot \|\nabla f_\theta(g(c))\|_2 + \frac{1}{2}M\alpha. \end{aligned} \tag{16}$$

Therefore, for $\forall d, \forall \hat{d}$, The following inequality holds

$$|P(y|d;\theta) - P(y|\hat{d};\theta)| < 2\sqrt{u} \cdot \alpha \cdot \|\nabla f_\theta(g(c))\|_2 + M\alpha. \tag{17}$$

For the gradient update of $f_\theta$, we can decompose it into the sum of two parts:

$$\frac{\partial L}{\partial \theta} = \frac{\partial L_\mathcal{X}}{\partial \theta} + \frac{\partial L_{\mathcal{X}'}}{\partial \theta}. \tag{18}$$

Since $N_1 \gg N_2 \rightarrow \frac{\partial L_\mathcal{X}}{\partial \theta} \gg \frac{\partial L_{\mathcal{X}'}}{\partial \theta}$, $f_\theta$ will prioritize ensuring convergence for $\mathcal{X}$. Therefore, we have $P(y|d;\theta) > P(y|\hat{d};\theta)$.

Substituting it into Eq. (17), we have

$$P(y|\hat{d};\theta) > P(y|d;\theta) - 2\sqrt{u} \cdot \alpha \cdot \|\nabla f_\theta(g(c))\|_2 - M\alpha. \tag{19}$$

$\square$

## B.3. Proof of Upper Bound for Hessian Matrix (cont. Lemma 1)

*Proof.* We are going to prove that the spectral norm of the Hessian matrix mentioned in Lemma 1 has an upper bound, i.e., $\|Hf_\theta(\xi)\|_2 \leq M, \forall \xi \in [g(d), g(d')]$ holds, take a fully connected network as an example:

Let the classifier $f_\theta$ be an L-layer fully connected neural network with parameters $\theta = W_1, b_1, ..., W_L, b_L$ where the input is $x \in \mathbb{R}^{dim}$ and the output is $y \in \mathbb{R}^K$. The computation at each layer is defined as follows:

$$z_l = W_l \sigma(z_{l-1}) + b_l, \tag{20}$$

where $\sigma$ is the activation function and $z_0 = x$ and the norm of the input $x$ is bounded by $B$.

Here, popular activation functions such as Sigmoid, Swish, and GLUE are twice differentiable with bounded second derivatives. Furthermore, the weight matrix of each layer satisfies $\|W_l\|_2 \leq C$, where $C$ is a constant.

For the Hessian matrix $H^k$, which is the k-th component of the output $f_\theta^k(x)$ can be expanded as:

$$H^k = \sum_{l=1}^{L} \frac{\partial^2 f_\theta^k}{\partial W_l^2} + \text{Cross Terms}, \tag{21}$$

To simplify the analysis, we focus on the diagonal blocks and ignore the cross terms, as their norms are typically much smaller than those of the main diagonal blocks.

Therefore, for the l-th layer, we have:

$$\begin{aligned} H_l^k &= \frac{\partial^2 f_\theta^k}{\partial W_l^2} = \frac{\partial}{\partial W_l}\left(\frac{\partial f_\theta^k}{\partial W_l}\right) \\ &\Rightarrow \|H_l^k\|_2 \leq K \cdot \|a_{l-1}\|_2^2 \cdot \|W_l\|_2^2. \end{aligned} \tag{22}$$

Furthermore, with the condition $\|W_l\|_2^2 \le C_l$ and $\|x\|_2 \le B$, we obtain the following inequality using induction:

$$\|a_{l-1}\|_2 \le B \cdot \prod_{i=1}^{l-1} C_i \|\sigma'\|_\infty$$

$$\Rightarrow \|H_l^k\|_2 \le K \cdot B^2 \cdot C_l^2 (\prod_{i=1}^{l-1} C_i^2 \|\sigma'\|_\infty^2). \tag{23}$$

The spectral norm of the Hessian matrix of $f_\theta$ can be obtained by summing the norms of the Hessian blocks from each layer:

$$\|Hf_\theta(\xi)\|_2 \le \sum_{l=1}^{L} \|H_l^k\|_2$$

$$\Rightarrow \|Hf_\theta(\xi)\|_2 \le K \cdot B^2 \cdot \sum_{l=1}^{L} (C_l^2 \prod_{i=1}^{l-1} C_i^2 \|\sigma'\|_\infty^2). \tag{24}$$

$\square$

## C. Why Our Theory Makes Sense to Demonstrate Bias Existence?

### C.1. Lemma 3.1

This lemma is useful in demonstrating the existence of semantic bias in text and images because it provides a mathematical bound on how small changes in the input space (according to some function $g$) translate into changes in the classifier's output. The key insight is that if two semantically similar inputs (i.e., ones mapped close to each other by $g$) result in significantly different outputs, it suggests that the classifier is sensitive to specific aspects of the input that may not align with human perception of similarity—indicating bias.

**Why This Lemma Makes Sense in Demonstrating Bias.**

(1) *Local Sensitivity to Features:* The lemma highlights how changes in input embeddings $g(d)$ and $g(d')$ propagate through the classifier. If the upper bound is large for semantically similar inputs, the classifier may be over-sensitive to subtle, potentially biased variations.

(2) *Dependence on Gradient and Curvature:* The bound depends on $\|\nabla f_\theta(g(d))\|_2$ and the Hessian norm $M$, meaning that sharp decision boundaries (high curvature) and large gradient norms can amplify small differences in representation space, possibly leading to biased decisions.

(3) *Quantifying Semantic Bias:* If semantic similarity (low $\|g(d) - g(d')\|$ does not guarantee a correspondingly small change in the classifier's output, this suggests that the classifier does not treat semantically similar inputs equivalently—an indication of bias.

### C.2. Lemma 3.2

This lemma provides a theoretical justification for how local label imbalances lead to biased classifier behavior, reinforcing the existence of semantic bias in text and images. It highlights the mechanism by which majority label dominance affects classification decisions within a local region of the representation space.

**Why This Lemma Makes Sense in Demonstrating Bias.**

(1) *Local Majority Influence:* If one label dominates in a local region of the embedding space $B(c, \alpha)$, empirical risk minimization (ERM) encourages the classifier to favor that majority label, leading to biased predictions.

(2) *Persistent Misclassification of the Minority Class:* The second claim establishes a lower bound on the classifier's misclassification probability for the minority class, demonstrating that bias is not incidental—it is structurally inevitable due to the local label imbalance.

(3) *Worsening Over Time:* The third claim states that this misclassification bias increases during training, suggesting that deeper training exacerbates bias rather than mitigating it. This aligns with real-world observations where models tend to overfit dominant patterns in the data.

**How This Relates to Semantic Bias.**

(1) If a classifier systematically favors majority-labeled instances, it implies that certain semantics (e.g., particular words, styles, or features) are overprivileged, while others are suppressed.

(2) This is particularly relevant in applications like natural language processing and computer vision, where spurious correlations (e.g., associating specific words with certain sentiments or skin tones with particular classifications) arise due to dataset biases.

# D. Supplementary Experiments

### D.1. Precision-Recall Curve

To further demonstrate the effectiveness of SCISSOR, we plot the precision-recall (PR) curves on the 4 datasets used, as shown in Fig. 4.

We observe that in all cases, SCISSOR consistently shows a higher Area Under the Curve (AUC) than the baselines. In the CV domain, the largest improvement comes from the ViT and DINOv2 models on the Chest-XRay dataset. As for NLP datasets, the most significant difference is observed with the BERT and RoBERTa models on the Yelp dataset. For LLaMA3, the improvement of SCISSOR is relatively slight, whereas we can still observe that our PR curve consistently lies above that of the baseline.

| Model | BERT | | RoBERTa | | LLaMA3 | |
|---|---|---|---|---|---|---|
| | Intra-cluster | Inter-cluster | Intra-cluster | Inter-cluster | Intra-cluster | Inter-Cluster |
| GYAFC | 3.06 * | 1.87 * | 2.53 * | 2.26 * | 2.34 * | 2.27 * |
| Yelp | 11.35 * | 6.80 * | 11.56 * | 6.50 * | 7.12 * | 6.99 * |

*Table 11.* Jaccard coefficient of words. For clear comparison, we increased the value by 100 times. * indicates a statistically significant difference between the Intra-cluster and Inter-cluster groups, with a p-value less than 0.0001.

| Model | BERT | | RoBERTa | | LLaMA3 | |
|---|---|---|---|---|---|---|
| | Intra-cluster | Inter-cluster | Intra-cluster | Inter-cluster | Intra-cluster | Inter-Cluster |
| GYAFC | 25.28 * | 23.38 * | 24.01 * | 23.34 * | 23.51 | 23.53 |
| Yelp | 27.26 * | 24.78 * | 27.75 * | 24.78 * | 24.98 | 25.11 |

*Table 12.* Levenshtin similarity between sequence of words. For clear comparison, we increased the value by 100 times. * indicates a statistically significant difference between the Intra-cluster and Inter-cluster groups, with a p-value less than 0.0001.

### D.2. Principal Component Analysis

To visually demonstrate how SCISSOR remaps the semantic space, we selected the RoBERTa model with the greatest performance improvement in the experiment, along with the Yelp dataset, for testing. We applied Principal Component Analysis (PCA) to reduce the outputs of both the debiased module and the original RoBERTa outputs to two dimensions, as shown in Figure 5.

### D.3. The Relationship Between Semantic Space Clustering and Linguistic Features

As an additional part, we propose two hypotheses specifically for the embeddings generated by language models:

**Samples within the same cluster may share more similar word compositions.** To verify this, we conducted 100,000 random sampling trials and calculated the Jaccard similarity coefficients for intra-cluster and inter-cluster pairs. The results are presented in Table 11.

**Samples within the same cluster may exhibit higher word string matching.** We performed 100,000 random sampling trials and computed Levenshtein similarity scores to evaluate the differences between randomly selected intra-cluster and inter-cluster pairs. The results are shown in Table 12.

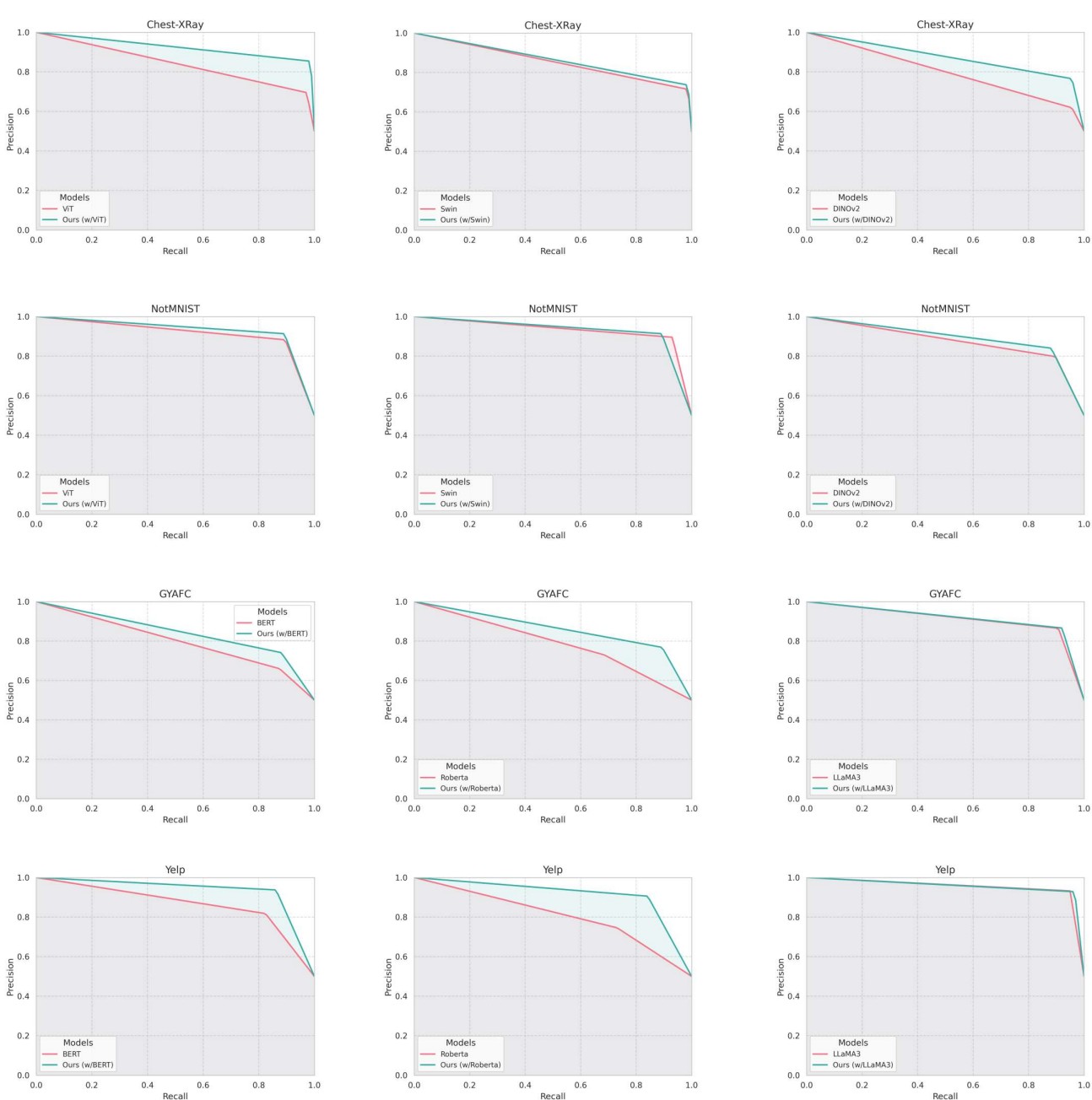

*Figure 4.* Precision-Recall curve.

We observe a strong correlation between semantic clusters and both word overlap and string matching, as shown in Tables 11 and 12. Samples within the same cluster (intra-cluster) exhibit a significantly higher Jaccard index compared to those in different clusters (inter-cluster). In particular, for embeddings from the BERT model, the Jaccard index for intra-cluster pairs is approximately 70% higher than that for inter-cluster pairs.

Similarly, intra-cluster pairs for BERT and RoBERTa embeddings exhibit significantly higher Levenshtein similarity than inter-cluster pairs. However, for the more powerful model LLaMA, the differences between the two groups are not statistically significant.

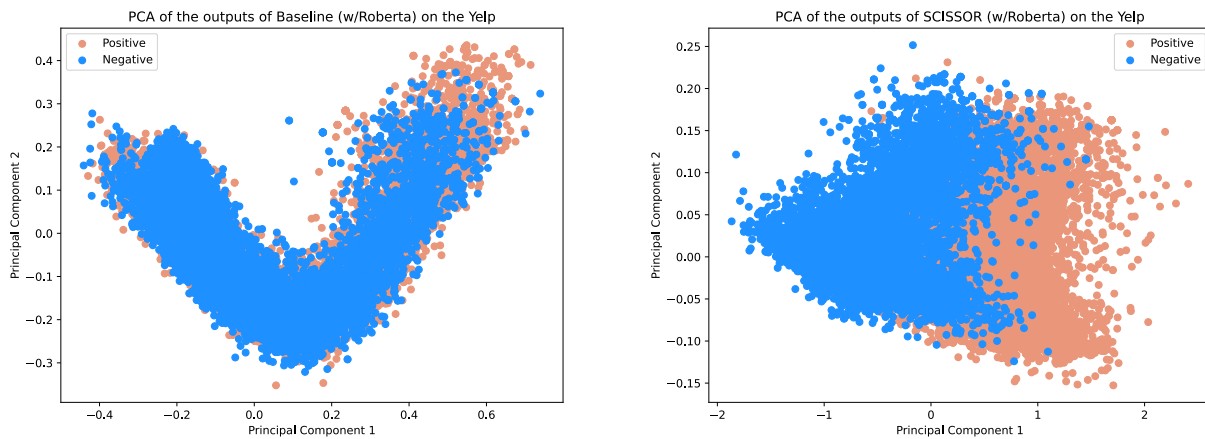

*Figure 5.* The comparison of PCA results for positive and negative sample embeddings in the semantic space of SCISSOR(w/Roberta) and Roberta, with the experiment conducted on the Yelp training set.

## E. Scalability and Complexity of MCL

Although Markov clustering can incur a quadratic cost due to pairwise similarity computations, we underscore that it is performed *offline* on the training set–i.e., it does not need to be repeated once the clusters are established. In practice, especially for large datasets, one can mitigate the computational load by adopting approximate nearest-neighbor methods (e.g., FAISS (Johnson et al., 2017) or HNSW (Malkov & Yashunin, 2018)) to sparsify the initial similarity graph before running the Markov clustering. This significantly reduces both runtime and memory overhead without compromising the integrity of the clusters. Moreover, advanced parallelization strategies, such as block-partitioned similarity calculations on GPUs, can further scale our clustering to millions of samples. As a result, while the theoretical complexity is $O(n^2)$, well-engineered approximations allow Markov clustering to be applied efficiently in domains like vision and NLP where data can be extensive. Crucially, this one-time clustering does *not* impact inference-time latency; once the semantic clusters are identified, SCISSOR integrates seamlessly as a lightweight module on top of the frozen pretrained model.

