# OpenReview forum: "SCISSOR: Mitigating Semantic Bias through Cluster-Aware Siamese Networks for Robust Classification"
_ICML.cc/2025/Conference — ICML 2025 poster_

### Official Review · Reviewer_2fdf · 2025-02-28

**Overall Recommendation:** 4

**Summary:**

This paper presents a novel debiasing architecture that leverages siamese networks and clustering techniques to mitigate spurious correlations in learned embeddings. The proposed approach remaps the embedding space to discourage unwanted dependencies between inputs and outputs while preserving meaningful semantic clusters.
A key advantage of this method is that it eliminates the need for extensive data augmentation or text rewriting, making it a more efficient alternative for debiasing.

The architecture introduces a debiasing module designed to disrupt semantic shortcuts, which are often responsible for model biases. This module is integrated between a pre-trained model and the classification head.
Extensive experiments on both image and text datasets demonstrate the effectiveness of this approach, showcasing its ability to enhance generalization by reducing the impact of spurious correlations.

**Claims And Evidence:**

The authors have done a good job in presenting lemmas on the existence of semantic biases, deriving insightful theoretical observations, and validating their claims empirically. The combination of theoretical derivations and experiments strengthens their argument and provides evidence for their claims.

**Essential References Not Discussed:**

Not to the best of my knowledge.

**Experimental Designs Or Analyses:**

The experiments effectively demonstrate the performance improvements of SCISSOR across multiple benchmarks, showing significant gains over baselines. However, it would be valuable to evaluate alternative clustering techniques and compare their impact on model performance, particularly for larger architectures where clustering appears weaker.

**Methods And Evaluation Criteria:**

While the paper effectively presents its method, there are some concerns regarding the reliance on Markov clustering. If the clustering does not accurately reflect the semantic structure of the embeddings, it may fail to suppress shortcuts effectively. Exploring alternative clustering methods such as Spectral Clustering, DBSCAN, or Gaussian Mixture Models could enhance robustness, particularly for larger models.

While the authors claim that larger models exhibit a weaker cluster effect, as indicated by a high Hopkins statistic, it is worth considering that this may be due to the difficulty of identifying clusters in high-dimensional spaces rather than an inherent weakening of the clustering effect itself. This could be a reason why SCISSOR seems to work best on smaller architectures.

**Other Comments Or Suggestions:**

1. Typos:
    1. Line 127, left column: "A employs LLMs"
    2. Line 266 right column: "debasing" -> "debiasing"


2. What do the different background colors in the table in step2 of figure 2 represent?

**Other Strengths And Weaknesses:**

Strengths: The proposed model is novel, especially in how it remaps the latent space to suppress spurious clusters. It also forces the model to focus on real discriminative features rather than cluster membership, improving generalization.

Weaknesses: The reliance on pre-trained embeddings being well-structured at the start of training could limit the method’s applicability to scenarios where embeddings are noisy or uninformative.

**Questions For Authors:**

Have you considered alternative clustering techniques to improve performance on larger models?

**Relation To Broader Scientific Literature:**

This paper contributes to literature on spurious correlations and shortcut learning. The methodology aligns with recent work on debiasing techniques and latent space interventions.

**Theoretical Claims:**

The paper’s theoretical contributions are well-structured, particularly in defining semantic biases and how they influence shortcut learning.

I have skimmed the proof of the lemmas in the appendix.

---

> ### Author Rebuttal · Authors · 2025-03-28
>
> **1. On Clustering Method (with Larger Models)**
>
> We conducted an ablation experiment using DBSCAN with LLaMA3, the largest model we used. The results are as follows:
>
> | Dataset        | ACC (GYAFC) | F1 (GYAFC) |  | ACC (Yelp) | F1 (Yelp) |
> |---------------|------------|------------|--|------------|-----------|
> | LLaMA3         | 89.37       | 0.89       |  | 94.57      | 0.95      |
> | LLaMA3 (w/ DBSCAN)  | 89.91       | 0.90       |  | 95.41      | 0.95      |
>
> We observed that the improvement of DBSCAN and MCL on LLaMA3 is similar, both being approximately 1%. This indicates that SCISSOR exhibits robustness to variations in clustering algorithms.
>
> **2. On Semantic Bias and Initial Embeddings**
>
> Since most contemporary debiasing works rely on pre-trained language models, we followed their works. We will consider debiasing methods in future work that do not rely on initial information.
>
> **3. On Background Colors in the Table in Step2 of Figure 2**
>
> We are sorry for making the confusion. The background colors are used to distinguish samples (i.e., Anchor, Positive, Intermediate, Negative). Their function is same to the color of the "Sun" and "Moon" markers. We will improve the readability of Figure 2 in the camera-ready version.

---

> > ### Comment · Reviewer_2fdf · 2025-04-02
> >
> > Thanks for you rebuttal, I will keep my accept score

---

### Official Review · Reviewer_eygs · 2025-03-13

**Overall Recommendation:** 1

**Summary:**

## Summary
This work aims at learning an adapter on pretrained representation to "filter out classification-irrelevant semantic features", to help out-of-distributation robustness. The author proposes a compliciated approach incorporating clustering, reweighting, contrastive learning, and creating “intermediate sample".



## Strongthness
- Out-of-distributation generalization is a challenging and interesting topic.

## Weakness

- This work claims " first to identify and demonstrate, both theoretically and empirically, that imbalances in the semantic distribution of samples can also lead to the shortcut problem." However, imbalance and shortcut learning is not new. Theorical work includes [1]. Experimental works includes GroupDRO and [3].

Please note that "semantic distribution" notion in this paper doesn't make the claim novel. Because one can treat the "semantic embedding", i.e. the representation of pretrained modek, as a preprocessed input. And apply all the imbalance techniques in the literature.

- The proposed method is complicated and hard to undertsand. Maybe an Algorithm table help. There are many unclear parts in the method, including how to choose intermediate example? How to support dataset with multiple classes?

- Lack of experimental comparision between naive baselines, such as simple reweighting, IRM [2] and many variations. Table 2 and Table 3 are not very helpful.


[1] Chaudhuri, K., Ahuja, K., Arjovsky, M. &amp; Lopez-Paz, D.. (2023). Why does Throwing Away Data Improve Worst-Group Error?. Proceedings of the 40th International Conference on Machine Learning

[2] Arjovsky, M., Bottou, L., Gulrajani, I., & Lopez-Paz, D. (2019). Invariant risk minimization. arXiv preprint arXiv:1907.02893.

[3] Kirichenko, P., Izmailov, P., & Wilson, A. G. (2022). Last layer re-training is sufficient for robustness to spurious correlations. arXiv preprint arXiv:2204.02937.

**Claims And Evidence:**

check summary

**Essential References Not Discussed:**

check summary

**Experimental Designs Or Analyses:**

check summary

**Methods And Evaluation Criteria:**

check summary

**Other Comments Or Suggestions:**

check summary

**Other Strengths And Weaknesses:**

check summary

**Questions For Authors:**

check summary

**Relation To Broader Scientific Literature:**

check summary

**Theoretical Claims:**

check summary

---

> ### Author Rebuttal · Authors · 2025-03-28
>
> **1. On the Lack of Novelty**
>
> We respectfully clarify that, to our knowledge, shortcut learning stemming from a semantic imbalance in pre-trained models (which requires unsupervised analysis) is not widely studied. The theoretical reference [1] you mentioned deals with label quantity bias, rather than semantic distribution. In fact, our experiments (lines 192-196) use balanced label distributions. Likewise, GroupDRO also focuses on imbalances in categories, not the distribution of semantic features. The debiasing in our paper refers to mitigating shortcuts which universally across all groups in the training set, rather than focusing on the worst group. Meanwhile, [2] requires prior knowledge of shortcuts and access to an unbiased dataset, both of which can be impractical in real-world scenarios [3][4][5].
>
> Consequently, our assertion that “an imbalance in semantic distribution can lead to shortcuts” remains valid and distinct from the biases explored in [1], GroupDRO, and [2].
>
> **2. On Baseline Comparisons**
>
> As noted in our Introduction, existing shortcut-mitigation research typically addresses label imbalance or focuses on token/pixel-level biases. Please note that direct reweighting methods, for example, are not applicable here because our training samples are already balanced by label. This underscores our key statement: “Shortcuts can persist even when labels are balanced.”
>
> To address concerns about baselines, we have added a comparative experiment with IRM; please see our “to reviewer gyYJ” response for details. We also compare our approach with RAZOR, a leading method that already surpasses several simpler baselines. Because our method outperforms RAZOR, we did not repeat comparisons with those baselines, especially since their focus is on token-level biases, whereas our work targets semantic-level biases.
>
> **3. On the Usefulness of Tables 2 and 3**
>
> In response to concerns about the value of Tables 2 and 3, our main motivation arises from the imbalance in semantic distribution, which the Hopkins Statistic measures effectively. By comparing Tables 4 and 5, we illustrate that our method’s effectiveness increases as semantic imbalance grows, demonstrating that our semantic debiasing approach genuinely shifts how samples are represented.
>
> **4. Clarity Concerns**
>
> **4.1 Choosing Intermediate Samples**
>
> In lines 214-215, we explain that “Intermediate samples share the same label and cluster as the anchor,” and this process is also illustrated in Figure 2. Nevertheless, we will try to better highlight this aspect in our paper.
>
> **4.2 Multi-Label Classification**
>
> Our “Problem Formulation” section does not restrict us to binary classification. Figure 1 uses a binary example solely for illustrative purposes. In fact, we have already evaluated multi-label scenarios in the paper, such as the Not-MNIST dataset, which has ten labels (A-J) as noted in lines 291-293.
> Finally, we emphasize that shortcut learning is multifaceted; recent works on style-based shortcuts [6] and in-context learning shortcuts [7] highlight various dimensions of this issue. We believe their contributions are valuable, as they address different forms of shortcuts that are similar to our notion of semantic shortcuts.
>
> Finally, we emphasize that our work tackles semantic shortcuts, a fundamentally different challenge than label-based or count-based bias mitigation. Addressing these subtler cues demands deeper scrutiny than simply balancing label distributions or removing obvious spurious correlations. Consequently, our approach necessitates unsupervised techniques to effectively uncover and mitigate these hidden biases.
>
> [1] Chaudhuri, K., Ahuja, K., Arjovsky, M. & Lopez-Paz, D. (2023). Why does Throwing Away Data Improve Worst-Group Error? Proceedings of the 40th International Conference on Machine Learning
>
> [2] Kirichenko, P., Izmailov, P., & Wilson, A. G. (2022). Last layer re-training is sufficient for robustness to spurious correlations. arXiv preprint arXiv:2204.02937.
>
> [3] Weizhi Xu, Qiang Liu, Shu Wu, Liang Wang (2023), Counterfactual Debiasing for Fact Verification
>
> [4] Shuo Yang, Bardh Prenkaj, Gjergji Kasneci. (2025). RAZOR: Sharpening Knowledge by Cutting Bias with Unsupervised Text Rewriting
>
> [5] Zeming Chen, Qiyue Gao, Antoine Bosselut, Ashish Sabharwal, Kyle Richardson. (2024). DISCO: Distilling Counterfactuals with Large Language Models
>
> [6] Yuqing Zhou, Ruixiang Tang, Ziyu Yao, Ziwei Zhu. (2024). Navigating the Shortcut Maze: A Comprehensive Analysis of Shortcut Learning in Text Classification by Language Models
>
> [7] Joonwon Jang, Sanghwan Jang, Wonbin Kweon, Minjin Jeon, Hwanjo Yu. (2024). Rectifying Demonstration Shortcut in In-Context Learning

---

### Official Review · Reviewer_347M · 2025-03-14

**Overall Recommendation:** 3

**Summary:**

This paper proposes SCISSOR, a debiasing approach that mitigates semantic biases in classifiers by disrupting semantic clusters that create shortcut learning. Using a Siamese network with Markov Clustering, it creates contrastive learning pairs to remap the semantic space, and through experiments, showed strong improvements across six models and four datasets.

**Claims And Evidence:**

The claims made in the submission are supported by clear and convincing evidence.

**Essential References Not Discussed:**

N/A

**Experimental Designs Or Analyses:**

The experimental design is well-structured, which includes comparison with related methods across a wide range of benchmarks and backbones. Multiple metrics are reported, and a computational efficiency analysis is also included.

**Methods And Evaluation Criteria:**

The benchmark datasets and model selection cover vision and language, making the results broadly applicable. The evaluation criteria align well with the problem of semantic bias mitigation.

**Other Comments Or Suggestions:**

N/A

**Other Strengths And Weaknesses:**

Please see sections above.

**Questions For Authors:**

N/A

**Relation To Broader Scientific Literature:**

In comparison to previous works, this work is motivated by the fact that balanced data can still have semantic biases. The proposed method is instead designed to remap the embedding space itself, complementing both dataset-centric and DRO-based methods by specifically targeting label-skewed clusters.

**Theoretical Claims:**

The paper outlines two lemmas that explain how semantic clusters affect classification. Formal proofs are also included in the appendix.

---

> ### Author Rebuttal · Authors · 2025-03-28
>
> Thank you for your positive feedback on our work. We sincerely appreciate your time and support. We'll address any further questions you should have during the discussion period to improve our paper, and make it through the finish line.

---

### Official Review · Reviewer_gyYJ · 2025-03-21

**Overall Recommendation:** 2

**Summary:**

This work introduces SCISSOR (Semantic Cluster Intervention for Suppressing Shortcut), a Siamese network-based debiasing approach that remaps the semantic space by discouraging latent clusters exploited as shortcuts. Shortcut learning is a critical issue that undermines model generalization to out-of-distribution data. Through extensive evaluation on various models and benchmarks, SCISSOR demonstrates its effectiveness in mitigating shortcut learning and promoting more robust machine learning models.

**Claims And Evidence:**

The paper’s main claim that semantic bias can be mitigated through cluster-aware Siamese networks, is grounded in both theoretical observations and empirical experiments.

**Essential References Not Discussed:**

Mitigating shortcut learning and spurious correlations is fundamental to achieving robust machine learning. Existing literature (for example [1, 2]) has introduced various strategies to address these challenges. A thorough, in-depth comparison with such works is necessary to highlight the position of the proposed approach in the broader research field.

[1] Invariant Risk Minimization. 2019.

[2] Distributionally Robust Neural Networks for Group Shifts: On the Importance of Regularization for Worst-Case Generalization. 2020.

**Experimental Designs Or Analyses:**

The experimental designs involve six models, and the results and analysis demonstrate both the validation of semantic shortcuts and the effectiveness of the proposed method.

**Methods And Evaluation Criteria:**

The semantic cluster intervention makes sense for suppressing shortcuts, with its performance evaluated using accuracy and F1 metrics on multiple datasets that are suitable for this assessment.

**Other Comments Or Suggestions:**

None

**Other Strengths And Weaknesses:**

Advantages

1. This work addresses the critical issue of shortcut learning in robust machine learning, which is an important and challenging research area.

2. The proposed method appears reasonable and is supported by experimental details that demonstrate its effectiveness.

Weaknesses

1. The theoretical analysis primarily focuses on identifying biases and discussing related observations. A more in-depth and stylized analysis of the cluster-aware Siamese network for robust classification would further strengthen the theoretical foundation.

2. Figure 1 illustrates sentiment classification, but more visualizations using real data would help showcase the semantic space and better demonstrate the proposed approach’s effectiveness.

3. A comprehensive evaluation on large-scale, real-world datasets (such as [1, 2]) would provide stronger evidence of the method’s practical applicability.

[1] WILDS: A Benchmark of in-the-Wild Distribution Shifts. ICML 2021.

[2] In Search of Lost Domain Generalization. 2020.

**Questions For Authors:**

Refer to the Weaknesses section for more detailed questions.

**Relation To Broader Scientific Literature:**

This work is closely related to shortcut learning for generalization under distribution shifts, biases, and spurious correlations issues that are critical in the field of robust machine learning.

**Theoretical Claims:**

I reviewed the lemma discussing the existence of semantic bias and the theory-grounded observations, and they appear logically sound and correct to me.

---

> ### Author Rebuttal · Authors · 2025-03-28
>
> **1. We added a comparative experiment with Invariant Risk Minimization (IRM) [1] and will include the results as well the corresponding references in the Related Work section.**
>
> | Dataset        | ACC (GYAFC) | F1 (GYAFC) |  | ACC (Yelp) | F1 (Yelp) |
> |---------------|------------|------------|--|------------|-----------|
> | BERT           | 70.40       | 0.70       |  | 81.17      | 0.81      |
> | BERT (w/IRM)   | 77.82       | 0.77       |  | 87.29      | 0.87      |
> | BERT (w/Ours)   | 78.20       | 0.78       |  | 90.65      | 0.91      |
> | RoBERTa        | 73.66       | 0.73       |  | 75.76      | 0.76      |
> | RoBERTa (w/IRM) | 79.61       | 0.79       |  | 84.53      | 0.84      |
> | RoBERTa (w/Ours) | 81.34       | 0.81       |  | 87.79      | 0.88      |
> | LLaMA          | 89.40       | 0.89       |  | 95.00      | 0.95      |
> | LLaMA (w/IRM)  | 83.65       | 0.83       |  | 94.87      | 0.95      |
> | LLaMA (w/Ours)  | 89.46       | 0.89       |  | 95.20      | 0.95      |
>
> |Dataset|ACC (Chest-XRay)|F1 (Chest-XRay)| |ACC (Not-MNIST)|F1 (Not-MNIST)|
> |---------------|------------|------------|--|------------|-----------|
> | ViT           | 72.38       | 0.72       |  | 88.87      | 0.89      |
> | ViT (w/IRM)   | 80.47       | 0.80       |  | 89.37      | 0.89      |
> | ViT (w/Ours)   | 83.92       | 0.84       |  | 90.89      | 0.91      |
> | SWIN        | 84.54       | 0.84       |  | 92.72      | 0.93      |
> | SWIN (w/IRM) | 76.92       | 0.76       |  | 92.12      | 0.92      |
> | SWIN (w/Ours) | 88.65       | 0.89       |  | 92.74      | 0.93      |
> | DINOv2          | 68.94       | 0.66       |  | 85.40      | 0.85      |
> | DINOv2 (w/IRM)  | 69.64       | 0.67       |  | 85.39      | 0.85      |
> | DINOv2 (w/Ours)  | 73.59       | 0.72       |  | 85.75      | 0.86      |
>
> We observe that although IRM does mitigate shortcuts in many cases, our method still significantly outperforms it across all tests. Moreover, IRM performs worse than the baseline on small datasets, such as Chest-XRay (w/SWIN) and GYAFC (w/LLaMA). We attribute this to IRM assigning excessive training weight to features that remain invariant, which prevents other useful features from being accurately identified and utilized.
>
>
> **2. Adding a Visual Example of Semantic Bias**
>
> We used PCA to reduce the BERT embeddings of the Yelp dataset to two dimensions and visualized three clusters, as shown in the Anonymous github we attached in the paper. In the image (https://anonymous.4open.science/r/SCISSOR-3F55/unb.svg), the colors represent cluster labels, while in the image (https://anonymous.4open.science/r/SCISSOR-3F55/unb_l.svg), colors indicate task labels (positive/negative).
>
> We observe that different clusters are located in distinct regions. Furthermore, the task labels within each cluster are imbalanced: in two of the three clusters, positive samples are more prevalent, while in the remaining cluster, negative samples dominate. Nevertheless, the total number of positive and negative samples across all three clusters is equal.
>
> In this scenario, negative samples belonging to clusters with more positive samples are theoretically more likely to be misclassified as positive by the models (as suggested by the Lemma 1 and Lemma 2), leading to the semantic shortcut we discuss.
>
> Furthermore, in the image (https://anonymous.4open.science/r/SCISSOR-3F55/pca_SCISSOR.svg), we show the semantic representations of samples after being remapped by our debiasing module. Through using SCISSOR, we observe that the task-irrelevant cluster information has been successfully removed. In contrast, classification information is preserved, naturally leading to a distinct classification boundary among the samples. To better illustrate the separability of the samples here, we visualize the three-dimensional PCA results of the output of our debiasing module (https://anonymous.4open.science/r/SCISSOR-3F55/pca_SCISSOR_3d.svg).
>
> **3. Experiments on Large-Scale benchmarks**
>
> Thank you for these valuable pointers. We are running experiments on the WILD and DomainBed benchmarks. However, due to the large scale of these experiments, we may not be able to obtain results within the rebuttal period (we might make it within the discussion period). Nevertheless, we emphasize that our current experiments already include 4 commonly used datasets, which we believe provide a solid basis for evaluating our approach and demonstrating its reliability.
>
> [1] Arjovsky, M., Bottou, L., Gulrajani, I., & Lopez-Paz, D. (2019). Invariant risk minimization. arXiv preprint arXiv:1907.02893.

---

### Decision · Program_Chairs · 2025-05-01

**Decision:**

Accept (poster)

**Comment:**

The paper proposes SCISSOR, which mitigates semantic bias in pre-trained models by remapping latent spaces through cluster-aware Siamese networks. The method discourages shortcut learning by disrupting imbalanced semantic clusters, without requiring data augmentation or rewriting. The method is evaluated across four benchmarks and six models in both vision and NLP domains.

The paper addresses an underexplored aspect of shortcut learning by studying semantic biases arising from latent space imbalances. The reviewers largely agree that the method is novel, well-motivated, and theoretically grounded. The experimental evaluation is in general thorough to show the consistent improvements over existing debiasing baselines such as RAZOR and IRM. Reviewers appreciated the plug-and-play design, the breadth of evaluation, and the evidence that SCISSOR works across modalities and model scales.

The main concerns come from reviewer eygs questioning the novelty and clarity of the method. The authors, however, did a good job providing detailed and convincing rebuttals, clarifying distinctions from prior works focused on label imbalance and showing that simple reweighting or IRM are insufficient for their setting. In addition, the authors addressed gyYJ’s concerns by adding comparisons to IRM, and provided additional visualizations to illustrate semantic bias. While the large-scale evaluations were not completed, the current results already demonstrate sufficient evidence on the effectiveness of the proposed method. Some reviewers noted the reliance on clustering and initial embeddings, but ablations show the method's robustness across choices.

Considering the above aspects, the AC recommends acceptance to ICML.